# Impact of Medical Specialties on Diagnostic and Therapeutic Management of Elderly Cancer Patients

**DOI:** 10.3390/geriatrics8030062

**Published:** 2023-06-01

**Authors:** Ludovic Lafaie, Anne-Françoise Chanelière-Sauvant, Nicolas Magné, Wafa Bouleftour, Fabien Tinquaut, Thomas Célarier, Laurent Bertoletti

**Affiliations:** 1Département de Gérontologie Clinique, CHU de Saint-Étienne, F-42055 Saint-Etienne, France; 2INSERM, UMR1059, Equipe Dysfonction Vasculaire et Hémostase, Université Jean-Monnet, F-42055 Saint-Etienne, France; 3Département de Radiothérapie, Institut Bergonié, F-33076 Bordeaux, France; 4Département d’Oncologie Médicale, CHU de Saint-Etienne, F-42055 Saint-Etienne, France; 5Département de Santé Publique, CHU de Saint-Etienne, F-42055 Saint-Etienne, France; 6Gérontopôle Auvergne Rhône-Alpes, F-42055 Saint-Etienne, France; 7Chaire Santé des Ainés, Université Jean Monnet, F-42055 Saint-Etienne, France; 8Service de Médecine Vasculaire et Thérapeutique, CHU de Saint-Etienne, F-42055 Saint-Etienne, France; 9INSERM, CIC-1408, CHU de Saint-Etienne, F-42055 Saint-Etienne, France

**Keywords:** geriatric oncology, elderly, cancer

## Abstract

The management (diagnostic and therapeutic) of cancer in the geriatric population involves a number of complex difficulties. The aim of this study was to assess the impact of a medical specialty on the diagnostic and therapeutic management of elderly cancer patients. Four clinical scenarios of cancer in the geriatric population, with a dedicated survey to gather information regarding each clinical case’s diagnostic and therapeutic approaches, as well as the different criteria influencing physicians’ therapeutic decisions, were exposed to geriatricians, oncologists, and radiotherapists in Saint-Etienne. The surveys were filled out by 13 geriatricians, 11 oncologists, and 7 radiotherapists. There was a homogeneity of responses regarding the confirmation of cancer diagnostics in the elderly. There were strong disparities (inter- and intra-specialties) for several clinical situations regarding the therapeutic management of cancer. There were significant disparities in terms of surgical management, the implementation of a chemotherapy protocol, and the adaptation of the chemotherapy dosage. Contrary to oncologists, who primarily consider the G8 and the Karnofsky score, geriatric autonomy scores and frailty with cognitive assessment were the key factors determining diagnostic/therapeutic therapy for geriatricians. These results raise important ethical questions, requiring specific studies in geriatric populations to provide the homogenous management of elderly patients with cancer.

## 1. Introduction

According to French and international epidemiological data, the incidence of cancer continues to increase with age. In France, cancer is the leading cause of death, surpassing circulatory system diseases [1]. The median age at cancer diagnosis is 68 years for men and 67 years for women [2]. In 2017, French cancer patients aged 65 and over represented 62.4% of estimated cancer cases of all ages combined [3]. For people aged 85 and over, 45,993 new cases of cancer were estimated to have been diagnosed, i.e., 11.5% of all diagnosed cancers (9.3% among men and 14% among women) [3]. This incidence is increasing due to the growth and aging of the population as a result of increased life expectancy in industrialized countries. Despite recent advancements in antineoplastic treatments, as well as the optimization of the decisional care of cancer in the elderly population, old age remains a criterion for non-presentation at a multidisciplinary consultation meeting, despite its mandatory nature [4].

This population does not benefit from the same opportunities for therapeutic access as a population of patients in another age group. In fact, several studies showed that advancing age is frequently a pretext for under-treatment and occasionally results in a refusal to provide curative treatment (such as simple excision surgery, for example) [5,6]. There is a delay in diagnosis, especially when it comes to severe comorbidities experienced by geriatric patients, investigations that can be challenging to perform, but also because of the few discriminatory attitudes of some clinicians towards the elderly population [7,8]. In addition, the general public’s use of cancer screening procedures does not particularly apply to the senior population. The functional reserves of multiple organs decline with aging, which is expected to affect the pharmacokinetics and pharmacodynamics of chemotherapeutic drugs [9,10]. Moreover, due to comorbidities and interaction with long-term medicines used for other concomitant diseases, a “standard” treatment can therefore result in fatal consequences.

For several years now, a reflection has been developing around the role of oncogeriatrics in the therapeutic management of elderly patients [11,12]. Due to the high incidence of cancer in the elderly, a collaboration between oncologists and geriatricians has been established to investigate frailty criteria before considering oncological treatment, with the validation of a geriatric screening tool (G8 score) [13,14]. During this consultation, the geriatrician evaluates the patient’s physiological age and any potential comorbidities [15]. Comprehensive geriatric assessment in oncology is an integral part of international recommendations [13,16]. By clarifying what is related to the clinical manifestations of frailty and what is related to cancer, geriatric assessment can predict treatment tolerance, duration of hospital stay, dependency, and survivals [17]. Geriatric-led interventions are associated with improved chemo tolerance, indicating a positive effect of comprehensive geriatric assessments [18,19,20].

The oncologist–geriatrician partnership is now well established in most health care institutions. However, questions concerning therapeutic management must sometimes be asked well before the discovery of cancer in elderly patients. Several medical specialties are concerned about the issue of cancer in the elderly (general practitioners, geriatricians, oncologists, radiotherapists, hematologists, surgeons, etc.). Depending on the medical specialties, the management of patients may vary, in particular because of the sensitivity of certain doctors, or their proactivity in complex management. The impact of the original clinical discipline on these considerations is not well investigated. Medical management may differ depending on the specialty of origin, and several questions arise: Should we look for a tumor process in certain elderly people? Should the diagnostic pattern be the same in young or elderly patients? Should the dosage of oncological treatments be reduced in elderly patients? Would cancer be looked for and treated the same way if the patient was followed by a geriatrician, a radiotherapist, or an oncologist? These are multiple questions for which there is currently no strict consensus. As a result, various questions arise regarding both the diagnostic and therapeutic management of cancer in this population.

Thus, the objective of this study is to analyze the impact of medical specialties on the diagnostic and therapeutic management of cancer in elderly patients.

## 2. Materials and Methods

This study was conducted at the Saint-Etienne University Hospital and the Lucien Neuwirth Institute of Cancerology of the Loire.

Four clinical scenarios that were based on actual events were produced. These clinical situations are summarized in Table 1. They cover the main cancers affecting the population in the territory.

The different clinical situations have been written according to the same plan:Patient’s age, gender, lifestyle, and home support.Medical information: allergies, medical history, comorbidities, and number of medications prescriptions.The cancer history with the different para-clinical examinations performed.A standardized geriatric assessment including: Karnofsky/ECOG performance status score, ADL/IADL score, MMSE, mini-GDS/GDS, assessment of neurosensory deficits, number of falls in the previous year, nutritional assessment, sleep/asthenia assessment, physical activity evaluation, muscle strength assessment, mobility assessment, motor performance assessment.G8 score.

The medical oncologists, radiotherapists, and geriatricians received the clinical scenarios.

For each clinical case, the practitioners were asked, according to a closed questionnaire:Question 1: Would you have carried out diagnostic for cancer in this patient (biopsies, complementary exams...); Does a diagnostic approach seem reasonable to you?Question 2: Would you perform surgery if it were necessary for this patient?Question 3: Would you agree to prescribe chemotherapy for this patient (if chemotherapy was indicated)?Question 4: Would you consider adjusting the dose of chemotherapy for this patient (if chemotherapy was indicated)?

The remaining two questions in the questionnaire examined the criteria influencing the diagnostic and therapeutic management according to each clinical situation.

Question 5: What are the criteria that would guide your diagnostic approach of this clinical case?Question 6: What are the criteria that would affect your therapeutic approach of this clinical situation?

The questionnaires were sent to participants on a specific date. Participants (oncologists, geriatricians, and radiotherapists) were given 2 weeks to send in their responses. The data were then aggregated and analyzed accordingly.

All characteristics of the respondents (oncologists, geriatricians, and radiotherapists) were described by percentages. This study has a purely exploratory descriptive objective and is based on 4 clinical cases. Hence, the statistical analysis is mainly descriptive. The respondents’ answers for each question were recorded by specialty, using numbers and percentages as well as graphs in order to evaluate the concordance between specialties. The number of clinical cases and raters did not allow for the calculation of Cohen’s Kappa coefficient to produce a concordance indicator.

## 3. Results

### 3.1. Respondent Characteristics

The characteristics of the respondents are described in Table 2. The surveys were answered by 13 geriatricians, 11 oncologists, and 7 radiotherapists for each clinical case. All participants were hospital practitioners. Among the respondents, the response of each head of department was included.

### 3.2. Analysis Clinical Cases

The different clinical cases are presented in Table 1.

Clinical case 1 describes an 84-year-old married woman with a multifocal infiltrating ductal carcinoma of the right breast, who lives independently at home without any special assistance.

The second clinical case corresponds to a married 76-year-old patient, who receives important assistance at home from a nurse for the toilet and for taking his medicines. This patient has a tumor of the middle rectum, which, upon anatomopathological examination, was diagnosed as a well-differentiated Lieberkuhnian adenocarcinoma.

In clinical case 3, an 80-year-old widowed woman is described as having absolute autonomy. This patient has an adenocarcinoma of the pancreas with peritoneal carcinosis. 

Case report 4 presents a 78-year-old married patient who is a caregiver for his wife (who has Parkinson’s disease). The cancer is a bronchial adenocarcinoma.

Participants’ responses regarding the diagnostic and therapeutic management of cancer based on the clinical cases are described in Figure 1, Figure 2, Figure 3 and Figure 4.

### 3.3. Analysis of Question 5: What Are the Criteria That Would Guide Your Diagnostic Approach to This Clinical Case?

For geriatricians (*n* = 13), the criteria that most influenced their diagnostic management were 81% cognitive assessment, 75% patient’s ADL score, 73% comorbidities, 60% patient IADL score, nutritional assessment, and 56% patient mobility assessment. Gender, sleep and fatigue assessment, G8 score, home support, and depression assessment were the factors that had the least impact on their diagnostic management.

For oncologists (*n* = 11), the criteria that most influenced their diagnostic management were 75% comorbidities and cognitive assessment, 71% the Karnofsky/performance status score, and 50% antecedents. Gender, sleep and fatigue assessment, the number of medications on the patient’s initial prescription, motor performance assessment, and depression assessment were the factors that had the least impact on their diagnostic management.

Comorbidities (71%) and the Karnofsky/performance status score (68%) were the diagnostic management criteria that had the greatest impact on radiotherapists (*n* = 7).

The patient’s motor performance, mobility, muscle strength, physical activity, home support, depression assessment, gender, and the quantity of medications on the patient’s initial prescription were the characteristics that had the least impact on their diagnostic care.

The results are available in Table 3.

### 3.4. Analysis of Question 6: What Are the Criteria That Would Affect Your Therapeutic Approach to This Clinical Situation?

The results are available in Table 4.

The factors that influenced the geriatricians’ therapeutic management most were 87% comorbidities, 79% cognitive assessment, 77% nutritional assessment, 73% patient ADLs, 65% patient mobility assessment, 58% patient IADLs, 56% number of falls in the previous year, and 52% physical activity assessment. Gender, antecedents, sleep and fatigue assessment, depression evaluation, and the presence of a neurosensory deficiency were the factors that had the least impact on their therapeutic management.

For oncologists, the criteria most influencing their therapeutic management were 89% Karnofsky/performance status score, 86% comorbidities, 80% cognitive and nutritional assessment, 59% history, 55% age and home support, and 50% the number of falls in the previous year. The factors that had the least impact on their therapeutic management were gender, sleep and fatigue evaluation, depression evaluation, the number of drugs taken by the patient, and physical activity.

For radiotherapists, the criteria that most influenced their therapeutic management were comorbidities (100%), Karnofsky/performance status score (96%), antecedents (79%), G8 score (75%), cognitive and nutritional assessment (64%), and age (57%). The factors that had the least impact on their therapeutic management were their gender, depression assessment, motor performance assessment, muscular strength assessment, physical activity assessment, sleep and fatigue assessment and gender.

## 4. Discussion

This investigation demonstrated a homogeneity of responses regarding the diagnostics of cancer in the elderly for oncologists, radiotherapists as well as for geriatricians, whereas the therapeutic management was inhomogeneous through the three disciplines.

The diagnosis of cancer leads to homogeneous answers for the different clinical situations (Figure 1) and can raise several concerns about this diagnostic in an elderly cancer. Indeed, despite the invasive confirmation of cancer in certain patients, no treatment sanction would be proposed. Consequently, in order to avoid encountering situation of unreasonable obstinacy, the benefit–risk ratio of the examination allowing the diagnosis of cancer must be evaluated according to the patient’s characteristics. While keeping in mind that the patient’s wishes about the diagnostic for cancer must remain at the core of the diagnostic decision, these ethical considerations are crucial in daily practice.

Although diagnostic management seems to be homogeneous between the different medical specialties, this is not the case for therapeutic management. Indeed, regarding surgical management, there is a heterogeneity (Figure 2) in the management. The evaluation of the peri- and post-operative risk of elderly patients is complex. Operative mortality increases with age and surgical complexity [21]. There is no strict consensus on the surgical management of cancer in the elderly, and assessing the benefit–risk balance of surgery may be challenging (expected benefits, post-operative complications, etc.). A tool developed by the American College of Surgeons, called the Surgical Risk Calculator, can be used to estimate the risks of a surgical procedure [22]. The role of the geriatrician before and after oncological surgery appears to be essential both in terms of decision making and treatment, as demonstrated by a recent study in which the collaboration of the surgeon with a geriatrician before and after cancer surgery in elderly patients led to a significant reduction in post-surgical mortality [23].

Analyzing the answers given for question 3 and question 4 (Figure 3 and Figure 4), our results indicate inter- and intra-specialty heterogeneity for chemotherapy strategies. There is a major disparity in responses for all clinical cases, with uncertain chemotherapy protocols for each patient. These disparities in responses exist because of the lack of consensus on the therapeutic management of elderly cancer patients. This leads to totally different practices depending on the physician the patient encounters, particularly in terms of dose modifications for the first cycle of chemotherapy based on age. Oncologists may use their clinical judgment to reduce the dose of chemotherapy preemptively to reduce toxicity [24]. The rationale for these dose reductions is based on the existence of some recent studies that address chemotherapy dose reduction, including a phase III randomized clinical trial evaluating the optimal dose of oxaliplatin and capecitabine combination chemotherapy in advanced gastroesophageal cancer in frail, elderly patients.

This study found that patients who received a lower dose of chemotherapy had less toxicity and noninferior progression-free survival compared with patients treated with higher doses [25]. This lack of scientific information can only lead to discrepancies in management between different practitioners.

These differences in therapeutic management are indicative of the lack of specific data in the elderly cancer population. Medical practitioners rely on their own feelings, which may be biased by personal and professional experiences. This leads to this heterogeneity of medical practices according to the different clinical situations, whatever the medical specialty. This phenomenon was notably exposed in a 2009 study that looked at the differences in medical practices concerning the continuation (or not) of the prophylactic treatment of venous thromboembolic disease in palliative patients [26]. The absence of strictly documented recommendations leads to the rather random management of patients, depending on the practitioner.

The clinical situations presented in Table 1 are real-life situations, with therapeutic indications (chemotherapy, surgery, etc.). The four situations present patients who can receive chemotherapy with an ECOG performance status lower than 3. All individuals in the clinical situations maintained autonomy with the exception of clinical situation number 2 (ADL 3/6 IADL 0/4) in relation with neurocognitive disorders. The differences and similarities in the various clinical cases led to different sensitivities in the management of patients depending on the medical specialty. We can observe that the major influencing factors for geriatricians are frailty, with notable factors being the cognitive evaluation and the geriatric autonomy scores (ADL and IADL). This is unlike oncologists, who mainly consider the Karnofsky and the G8 onco-geriatric scores. This confirms how an oncologist and geriatrician team complement each other to help make good therapeutic decisions that will optimize oncological management [27,28].

This study has some weaknesses. This descriptive analysis remains low-powered, in particular because of the relatively limited number of respondents to the questionnaires, and the small number of clinical situations. It would seem relevant to obtain answers from general practitioners, and especially from surgeons, who are also important in the diagnostic and therapeutic management of cancer. Nevertheless, this descriptive analysis allows us to identify disparities in responses intra- and especially inter-specialty. Additionally, the investigation was carried out at an oncology center near a university hospital where a dedicated onco-geriatrics consultation team has existed for many years. Thereby, the extrapolability of the results may therefore be questioned. Consequently, it seems essential to pursue and initiate research projects in the field of geriatric oncology. All this will allow a better understanding of the adverse effects of cancer therapies in the field of geriatric oncology, but also to better evaluate their effectiveness in order to optimize cancer management.

Advanced age should not be a criterion of exclusion for the presentation of geriatric files in multidisciplinary consultation meetings. Despite the criterion of advanced age, the informed consent of the patient and his or her family must be taken into consideration. All the parameters of close collaboration between geriatricians and oncologists are the bulwark against the “under-treatment” and “over-treatment” of elderly cancer patients.

## Figures and Tables

**Figure 1 geriatrics-08-00062-f001:**
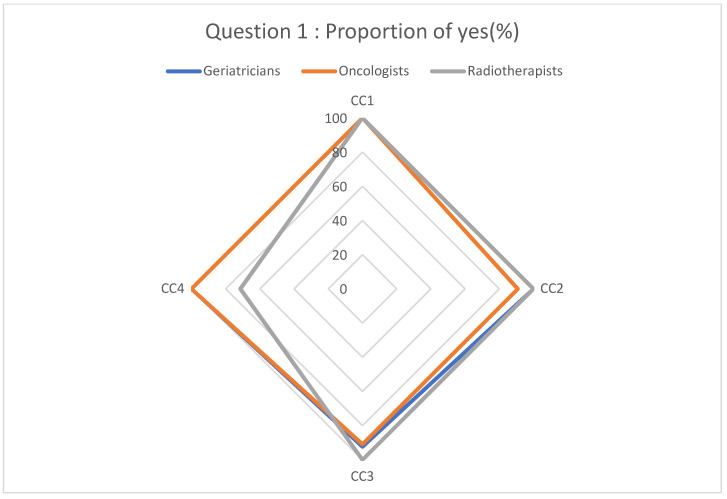
Proportion of “yes” responses to question 1.

**Figure 2 geriatrics-08-00062-f002:**
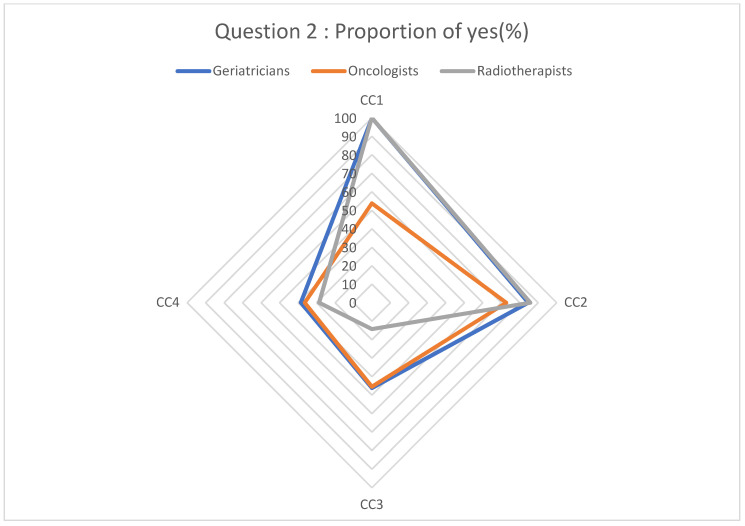
Proportion of “yes” responses to question 2.

**Figure 3 geriatrics-08-00062-f003:**
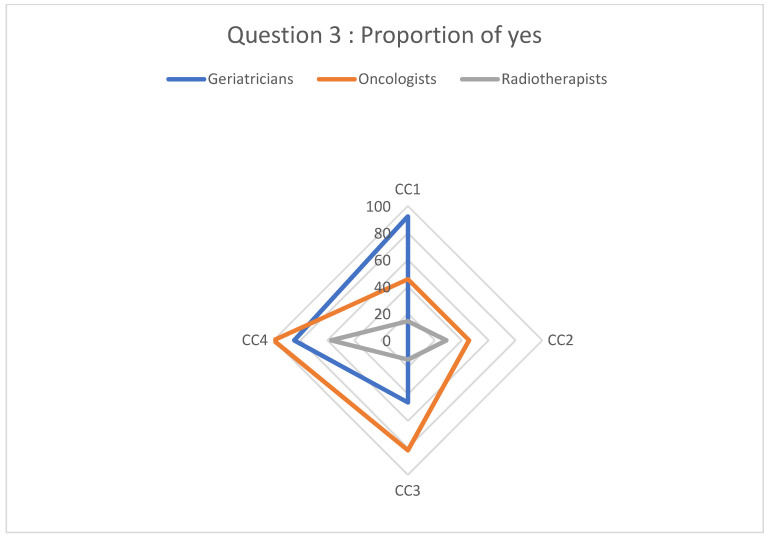
Proportion of “yes” responses to question 3.

**Figure 4 geriatrics-08-00062-f004:**
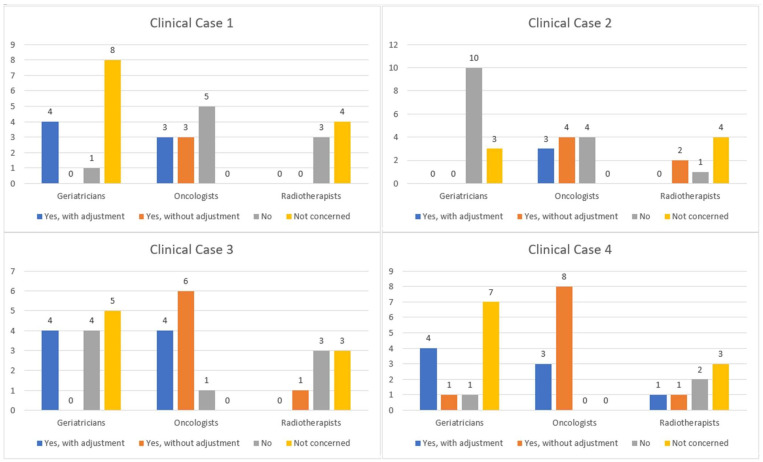
Analysis of question 4.

**Table 1 geriatrics-08-00062-t001:** Summary of clinical cases.

	Clinical Case 1	Clinical Case 2	Clinical Case 3	Clinical Case 4
Cancer	Multifocal infiltrating ductal carcinoma of the right breast, SBR III, RO+ 100%, RP+ 80%, Her2++ with negative FISH, Ki67 at 60%	Well-differentiated Lieberkuhnian adenocarcinoma of the middle rectum	Adenocarcinoma of the pancreas with peritoneal carcinosis	Locally advanced middle-lobe bronchial adenocarcinoma, stage 3A N2cT2b
Sex	Female	Male	Female	Male
Age	84	76	80	78
Technical aids	Cane	Rollator	None	None
Home care	None	Wireless caregiver pager Nurse Homemaker service	None	Homemaker service
Allergy	None	Penicillin	None	None
Toxics	2 glasses of alcohol per day	Tobacco 30PA	None	Tobacco 50 PA; 3 glasses of alcohol per day
Antecedents	Hysterectomy on fibroids	Bilateral total hip replacement Appendectomy	Left breast cancer treated in 1999 (mastectomy and radiotherapy) Right total hip replacement	Cholecystectomy Tonsillectomy
Comorbidities	Hypertension Gonarthrosis Anxiety and depression syndrome Age-related macular degeneration Chronic glaucoma Chronic renal failure Restrictive respiratory disorders	Obliterating arteriopathy of the lower limbs Anxiety and depression syndrome Alzheimer’s disease	Hypertension Migraines Rheumatoid arthritis Atrial fibrillation	Hypertension Hypothyroidism Chronic obstructive pulmonary disease
Number of medicines	7	2	10	3
Karnofsky(%)/ECOG performance status	K80-PS1	K50-PS2	K80-PS1	K90-PS1
ADL	6/6	3/6	6/6	6/6
IADL	2/4	0/4	4/4	3/4
MMS	27/30	18/30	26/30	24/30
Mini GDS–GDS	0/4–Not realized	3/4–12/15	0/4–Not realized	1/4–6/15
Neurosensory deficit	Blindness	None	Presbyopia + Myopia	Presbyopia + Presbycusis
Fall	None	3 in the last year	None	1 in the last year
Nutritional assessment: - BMI - Mini-MNA - MNA - Albumin	20.32 kg/m^2^ 10/14 18/30 33 g/L	23.66 kg/m^2^ 12/14 Not realized 38.2 g/L	22.03 kg/m^2^ 13/14 Not realized 39 g/L	24.4 kg/m^2^ 12/14 Not realized 38 g/L
Sleep	Disturbed	Correct	Correct	Correct
Physical activity	1.0 m/s	0.7 m/s	0.9 m/s	0.9 m/s
Muscular strength	Normal	Significantly reduced	Normal	Normal
Mobility: TUG	11 s	23 s	20 s	14 s
Motor performance	6/10	4/10	8/10	9/10
G8 score	8/17	7.5/17	12/17	10/17

**Table 2 geriatrics-08-00062-t002:** Characteristics of respondents.

	Geriatricians *n* = 13	Oncologists *n* = 11	Radiotherapists *n* = 7
-Gender:			
- Female	- 10 (77%)	- 5 (45%)	- 2 (29%)
- Male	- 3 (23%)	- 6 (55%)	- 5 (71%)
-Years of professional experience			
- >5 years	- 9 (69%)	- 8 (73%)	- 2 (29%)
- <5 years	- 4 (31%)	- 3 (27%)	- 5 (71%)

**Table 3 geriatrics-08-00062-t003:** Criteria influencing diagnostic approach according to specialty of origin (percentage of clinicians who checked the criteria).

Geriatricians	Oncologists	Radiotherapists
Cognitive assessment (81%)	Comorbidities (75%)	Comorbidities (71%)
ADL score (75%)	Cognitive assessment (75%)	Karnofsky and ECOG performance status scoring (68%)
Comorbidities (73%)	Karnofsky and ECOG performance status scoring (71%)	Antecedents (46%)
IADL score (60%)	Antecedents (50%)	G8 score (39%)
Nutritional assessment (60%)	Nutritional assessment (46%)	Age (32%)
Assessment of mobility (56%)	Age (39%)	Cognitive assessment (32%)
Physical activity assessment (42%)	G8 score (36%)	None (25%)
Age (39%)	Number of falls during the year (34%)	Nutritional assessment (21%)
Karnofsky and ECOG performance status scoring (31%)	Home care aids (25%)	Number of falls during the year (18%)
Assessment of motor performance (31%)	Assessment of muscle strength (23%)	ADL score (11%)
Number of falls during the year (29%)	Neurosensory deficit (18%)	IADL score (11%)
Assessment of muscle strength (27%)	None (18%)	Neurosensory deficit (4%)
Antecedents (14%)	IADL score (16%)	Sleep assessment (4%)
Number of medicines (14%)	Assessment of mobility (16%)	Sex (0%)
Neurosensory deficit (14%)	ADL score (14%)	Number of medicines (0%)
Mini-GDS/GDS (12%)	Physical activity assessment (14%)	Mini-GDS/GDS (0%)
Home care aids (12%)	Mini-GDS/GDS (11%)	Home care aids (0%)
None (12%)	Assessment of motor performance (9%)	Physical activity assessment (0%)
G8 score (8%)	Number of medicines (7%)	Assessment of muscle strength (0%)
Sleep assessment (8%)	Sleep assessment (5%)	Assessment of mobility (0%)
Sex (0%)	Sex (2%)	Assessment of motor performance (0%)

**Table 4 geriatrics-08-00062-t004:** Criteria influencing therapeutic approach according to specialty of origin (percentage of clinicians who checked the criteria).

Geriatricians	Oncologists	Radiotherapists
Comorbidities (87%)	Karnofsky and ECOG performance status scoring (89%)	Comorbidities (100%)
Cognitive assessment (79%)	Comorbidities (86%)	Karnofsky and ECOG performance status scoring (96%)
Nutritional assessment (77%)	Cognitive assessment (80%)	Antecedents (79%)
ADL score (73%)	Nutritional assessment (80%)	G8 score (75%)
Assessment of mobility (65%)	Antecedents (59%)	Cognitive assessment (64%)
IADL score (58%)	Age (55%)	Nutritional assessment (64%)
Number of falls during the year (56%)	Home care aids (55%)	Age (57%)
Physical activity assessment (52%)	Number of falls during the year (50%)	Home care aids (14%)
Karnofsky and ECOG performance status scoring (44%)	Assessment of mobility (43%)	Assessment of mobility (14%)
Assessment of motor performance (44%)	G8 score (41%)	IADL score (14%)
Age (42%)	IADL score (36%)	ADL score (14%)
Number of medicines (42%)	ADL score (34%)	Number of falls during the year (11%)
Home care aids (39%)	Assessment of motor performance (32%)	Neurosensory deficit (11%)
Assessment of muscle strength (37%)	Assessment of muscle strength (27%)	Number of medicines (11%)
G8 score (29%)	Neurosensory deficit (27%)	Mini-GDS/GDS (4%)
Neurosensory deficit (27%)	Physical activity assessment (25%)	Sex (4%)
Mini-GDS/GDS (23%)	Number of medicines (21%)	Assessment of motor performance (0%)
Sleep assessment (19%)	Mini-GDS / GDS (21%)	Assessment of muscle strength (0%)
Antecedents (12%)	Sleep assessment (9%)	Physical activity assessment (0%)
None (6%)	None (2%)	Sleep assessment (0%)
Sex (0%)	Sex (0%)	None (0%)

## Data Availability

All data are described in the paper.

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
