# Peer review of "Impact of Medical Specialties on Diagnostic and Therapeutic Management of Elderly Cancer Patients"

_geriatrics, 2023, doi:10.3390/geriatrics8030062_

Round 1
Reviewer 1 Report
Authors have given good introduction and background of research question raised during this article. There is a gap in knowledge regarding how different specialty will have effect on cancer patient treatment and this article is trying to shade some lights on. Clinical cases explained in tables has good enough details. Tables explaining criteria associated with decision is very well explained. very well written article.
Author Response
Many thanks to Reviewer 1 for his comments.
It is a real pleasure to read his comments.
Reviewer 2 Report
Dear authors
in your paper you evaluated different disciplines of medical specialists in their diagnostic / treatment choices in 4 different patients. You found differences between preferences of the groups. This is interesting and of relevance in the understanding for example the functioning of tumor boards.
My concerns are:
1. Why did you not include surgeons in the group? Especially as one question is about the indication of surgery. Why a radiotherapist? Please add surgeons and discuss.
2. The bottom line/information of your paper is: different subprofessions of medical doctors have different preferences. However other influencing factors might also be highly relevant: Gender, Age, years of professional experience, role in the department: intern, head of department etc..., working environment: private sector, hospital, outpatient service .Please provide more details about the different groups and discuss.
3. Your introduction is rather general and should be more focussed on your project: Why do you think it is necessary to investigate differences between different disciplines. Pleas rewrite. Again: Where is the surgeon?
4. Results: Graphs and original text give in several parts twice the same information: please shorten the text by referring to the graphs.
5. The discussion mainly repeats the results. However the discussion should include a an attempt to explain why doctors from different disciplines have different approaches to some aspects of oncologic patients.
6. This discussion should also include an interpretation of these differneces and discuss: why the findings are considered important for clinical decision making?
Author Response
Responses from the authors.
We are very happy to see that Reviewer 2 (R2) enjoyed the topic we studied in our work.
Please find attached the revisions made to the draft.
General comments
- Why did you not include surgeons in the group? Especially as one question is about the indication of surgery. Why a radiotherapist? Please add surgeons and discuss.
Responses from the authors :
We fully understand R2's point. Unfortunately the study has already been conducted, and it seems complex to add surgeons to the study. Moreover, in our French health system, the surgeon is exceptionally contacted directly in case of suspected cancer. Most of the time, a medical step is required before referral to a surgeon. However, since our objective was to evaluate medical responses, as stated as soon as the title, we did not include surgeons in the analysis. This may be a second work, endorsed by surgeons.
This is indeed a weakness of our study. Therefore we add this to the discussion:
“This study has some weaknesses. This descriptive analysis remains low-powered, in particular because of the relatively limited number of respondents to the questionnaires, and the small number of clinical situations. It would seem relevant to obtain answers from general practitioners, and especially from surgeons, who also play an essential role in the diagnostic and therapeutic management of cancer.”
- The bottom line/information of your paper is: different subprofessions of medical doctors have different preferences. However other influencing factors might also be highly relevant: Gender, Age, years of professional experience, role in the department: intern, head of department etc..., working environment: private sector, hospital, outpatient service. Please provide more details about the different groups and discuss.
Responses from the authors :
We fully agree with R2's point. As a result, the table has been changed to add the item "Years of professional experience".
Unfortunately, we did not collect the age of the participants. All participants work in a public hospital, not in the private sector. Until nowadays, there was no geriatricians in private practice "This study was conducted at the Saint-Etienne University Hospital and the Lucien Neuwirth Institute of Cancerology of the Loire".
We add the following sentence in the paragraph "3.1 Respondent Characteristics": “All participants, are hospital practitioners. Among the respondents, the response of each of the head of department is included.”
|
Geriatricians n = 13 |
Oncologists n = 11 |
Radiotherapists n = 7 |
|
-Gender : - Female - Male |
- 10 (77%) - 3 (23%) |
- 5 (45%) - 6 (55%) |
- 2 (29%) - 5 (71%) |
|
- Years of professional experience - > 5 years - < 5 years |
- 9 (69%) - 4 (31%) |
- 8 (73%) - 3 (27%) |
- 2 (29%) - 5 (71%) |
- Your introduction is rather general and should be more focussed on your project: Why do you think it is necessary to investigate differences between different disciplines. Pleas rewrite. Again: Where is the surgeon?
Responses from the authors :
We fully understand R2's point.
As a result, the introduction has been reworked and shortened; the paragraph concerning the medical specialty has been expanded and rewritten:
“The oncologist-geriatrician partnership is now well established in most health care institutions. However, questions concerning therapeutic management must sometimes be asked well before the discovery of cancer in elderly patients. Several medical specialties are concerned about the issue of cancer in the elderly (general practitioners, geriatricians, oncologists, radiotherapists, hematologists, surgeons, etc.). Depending on the medical specialties, the management of patients may vary, in particular because of the sensitivity of certain doctors, or their proactivity on complex management. The impact of the original clinical discipline on these considerations is not well investigated. Medical management may differ depending on the specialty of origin, and several questions arise: Should we look for a tumor process in certain elderly people? Should the diagnostic pattern be the same in young or elderly patients? Should the dosage of oncological treatments be reduced in elderly patients? Would cancer be sought and treated the same way if the patient is followed by a geriatrician, a radiotherapist, or an oncologist? These are multiple questions for which there is currently no strict consensus. As a results, various questions arise regarding both the diagnostic and therapeutic management of cancer in this population. »
Unfortunately, as mentioned in comment number 1, the study has already been conducted, and it seems complex to add surgeons to the study. “Moreover, in our French health system, the surgeon is exceptionally contacted directly in case of suspected cancer. Most of the time, a medical step is required before referral to a surgeon. However, since our objective was to evaluate medical responses, we did not include a surgeon.” This is indeed a weakness of our study. Therefore we add this to the discussion.
- Results: Graphs and original text give in several parts twice the same information: please shorten the text by referring to the graphs.
Responses from the authors :
We understand R2 comment. In consequence, we have shortened the text by referring to the graphs.
« 3.2. Analysis Clinical Cases
The different clinical cases are presented in Table 1.
Clinical case 1 describes an 84-year-old married, woman, with a multifocal infiltrating ductal carcinoma of the right breast who lives independently at home without any special assistance.
The second clinical case, corresponds to a married, 76-year-old patient, who receives an important assistance at home from a nurse for the realization of his toilet and for the taking of his medicines. This patient had a tumor of the middle rectum, which, upon anatomopathological examination was diagnosed as a well-differentiated Lieberkuhnian adenocarcinoma.
In clinical case 3, an 80-year-old widowed woman is described as having an absolute autonomy. This patient has an adenocarcinoma of the pancreas with peritoneal carcinosis.
Case report 4 presents a 78-year-old married patient who is a caregiver for his wife (who has Parkinson's disease). The cancer was a bronchial adenocarcinoma.
Participants' responses regarding the diagnostic and therapeutic management of cancer based on clinical cases are described in Figures 1, 2, 3, and 4.”
- The discussion mainly repeats the results. However the discussion should include a an attempt to explain why doctors from different disciplines have different approaches to some aspects of oncologic patients.
Responses from the authors :
We thank R2 for this comment, totally justified.
We have added in the discussion section :
“The clinical situations presented in Table 1 are real life situations, with therapeutic indications (chemotherapy, surgery...). The 4 situations present patients who can receive chemotherapy with an ECOG Performans Status lower than 3. All the clinical situations maintained an autonomy with the exception of clinical situation number 2 (ADL 3/6 IADL 0/4) in relation with neurocognitive disorders. The differences and similarities in the various clinical cases lead to different sensitivities in the management of patients depending on the medical specialty.”
- This discussion should also include an interpretation of these differences and discuss: why the findings are considered important for clinical decision making?
Responses from the authors :
We understand R2's point. Therefore, we have rewritten a large part of the discussion:
“This investigation demonstrated a homogeneity of responses regarding the diagnostic of cancer in the elderly for oncologists, radiotherapists as well as for geriatricians, whereas the therapeutic management was inhomogeneous through the three disciplines.
The diagnosis of cancer leads to homogeneous answers for the different clinical situations (Figure 1), and can raise several concerns about this diagnostic in an elderly cancer. Indeed, despite the invasive confirmation of cancer in certain patients, no treatment sanction would be proposed. Consequently, in order to avoid encountering situation of unreasonable obstinacy, the benefit-risk ratio of the examination allowing the diagnosis of cancer must be evaluated according to the patient’s characteristics. While keeping in mind that the patient's wishes about the diagnostic for cancer must remain at the core of the diagnostic decision, these ethical considerations are crucial in daily practice.
Although the diagnostic management seems to be homogeneous between the different medical specialties, this is not the case for the therapeutic management. Indeed, regarding surgical management, there was a heterogeneity (Figure 2) in the management. The evaluation of the peri- and post-operative risk of elderly patients is complex. Operative mortality increases with age and surgical complexity (23). There is no strict consensus on the surgical management of cancer in the elderly, and assessing the benefit-risk balance of surgery may be challenging (expected benefits, post-operative complications, etc.). A tool developed by the American College of Surgeons, called the Surgical Risk Calculator, can be used to estimate the risks for a surgical procedure (24). The role of the geriatrician before and after oncological surgery appears to be essential both in terms of decision-making and treatment, as demonstrated by a recent study in which the collaboration of the surgeon with a geriatrician before and after cancer surgery in elderly patients led to a significant reduction in post-surgical mortality (27).
Analyzing the answers given for question 3 and question 4 (Figure 3 and 4), our results indicate inter- and intra-specialty heterogeneity for chemotherapy strategies. There is a major disparity in responses for all clinical cases, with uncertain chemotherapy protocols for each patient. These disparities in responses exist because of the lack of consensus on the therapeutic management of elderly cancer patient. This leads to totally different practices depending on the physician the patient encounters, particularly on dose modifications for the first cycle of chemotherapy based on age. Oncologists may use their clinical judgment to reduce the dose of chemotherapy preemptively to reduce toxicity (28). The rationale for these dose reductions is based on the existence of some recent studies that address chemotherapy dose reduction, including a phase III randomized clinical trial evaluating the optimal dose of oxaliplatin and capecitabine combination chemotherapy in advanced gastroesophageal cancer in frail, elderly patients. This study found that patients who received a lower dose of chemotherapy had less toxicity and noninferior progression-free survival compared with patients treated with higher doses (32).This lack of scientific information can only lead to discrepancies in management between different practitioners.
These differences in therapeutic management are indicative of the lack of specific data in the elderly cancer population. Medical practitioners rely on their own feelings, which may be biased by personal and professional experiences. This leads to this heterogeneity of medical practices according to the different clinical situations, whatever the medical specialty. This phenomenon is notably exposed in a 2009 study that looked at the differences in medical practices concerning the continuation or not of prophylactic treatment of venous thromboembolic disease in palliative patients (Reference : Duverger C, Tardy B, Richard A, Célarier T, Bayle S, Cambou M, Perrot JL, Cathebras P, Gonthier R; Fédération de Soins Palliatifs de Saint Etienne. Traitement prophylactique de la maladie thromboembolique veineuse en soins palliatifs. Hétérogénéité des pratiques des médecins en réponse à 4 situations cliniques diffé-rentes [Prophylactic treatment of venous thromboembolic disease in palliative care. A survey about four different clinical cases]. Presse Med. 2009 Sep;38(9):1235-9.). The absence of strictly documented recommendations leads to a rather random management of patients, depending on the practitioner.
The clinical situations presented in Table 1 are real life situations, with therapeutic indications (chemotherapy, surgery...). The 4 situations present patients who can receive chemotherapy with an ECOG Performans Status lower than 3. All the clinical situations maintained an autonomy with the exception of clinical situation number 2 (ADL 3/6 IADL 0/4) in relation with neurocognitive disorders. The differences and similarities in the various clinical cases lead to different sensitivities in the management of patients depending on the medical specialty. We can observe that, the major influencing factors for geriatricians are frailty with notably the cognitive evaluation, and the geriatric autonomy scores (ADL and IADL). Unlike oncologists, who mainly consider the Karnofsky and the G8 onco-geriatric score. This confirms how an oncologist and geriatrician team complement each other to help making good therapeutic decisions that will optimize oncological management (35,36).
This study has some weaknesses. This descriptive analysis remains low-powered, in particular because of the relatively limited number of respondents to the questionnaires, and the small number of clinical situations. It would seem relevant to obtain answers from general practitioners, and especially from surgeons, who are also important in the diagnostic and therapeutic management of cancer. Nevertheless, this descriptive analysis allows us to identify disparities in responses intra and especially inter-specialty. Also, the investigation was done at an oncology center near a university hospital where a dedicated onco-geriatrics consultation has existed for many years. Thereby, the extrapolability of the results may therefore be questioned. Consequently, it seems essential to pursue and initi-ate research projects in the field of geriatric oncology. All this will allow a better under-standing of the adverse effects of cancer therapies in the field of geriatric oncology, but also to better evaluate their effectiveness in order to optimize cancer management.
Advanced age should not be a criterion of exclusion for the presentation of geriatric files in multidisciplinary consultation meetings. Despite the criterion of advanced age, the informed consent of the patient and his or her family must be taken into consideration. All the parameters of close collaboration between geriatricians and oncologists are the bul-wark against "under-treatment" and "over-treatment" of elderly cancer patients. »

Round 2
Reviewer 2 Report
Dear Authors thank you for considering our remarks. Best regards